# Interface engineering of Ta₃N₅ thin film photoanode for highly efficient photoelectrochemical water splitting

Jie Fu[1], Zeyu Fan[1,2], Mamiko Nakabayashi [3], Huanxin Ju [4], Nadiia Pastukhova[1], Yequan Xiao[1], Chao Feng[1], Naoya Shibata [3], Kazunari Domen [5,6] & Yanbo Li [1,2✉]

Interface engineering is a proven strategy to improve the efficiency of thin film semiconductor based solar energy conversion devices. Ta₃N₅ thin film photoanode is a promising candidate for photoelectrochemical (PEC) water splitting. Yet, a concerted effort to engineer both the bottom and top interfaces of Ta₃N₅ thin film photoanode is still lacking. Here, we employ n-type In:GaN and p-type Mg:GaN to modify the bottom and top interfaces of Ta₃N₅ thin film photoanode, respectively. The obtained In:GaN/Ta₃N₅/Mg:GaN heterojunction photoanode shows enhanced bulk carrier separation capability and better injection efficiency at photoanode/electrolyte interface, which lead to a record-high applied bias photon-to-current efficiency of 3.46% for Ta₃N₅-based photoanode. Furthermore, the roles of the In:GaN and Mg:GaN layers are distinguished through mechanistic studies. While the In:GaN layer contributes mainly to the enhanced bulk charge separation efficiency, the Mg:GaN layer improves the surface charge inject efficiency. This work demonstrates the crucial role of proper interface engineering for thin film-based photoanode in achieving efficient PEC water splitting.

[1] Institute of Fundamental and Frontier Sciences, University of Electronic Science and Technology of China, Chengdu, China. [2] Yangtza Delta Region Institute (Huzhou), University of Electronic Science and Technology of China, Huzhou, China. [3] Institute of Engineering Innovation, The University of Tokyo, Tokyo, Japan. [4] PHI China Analytical Laboratory, CoreTech Integrated Limited, Nanjing, China. [5] Office of University Professors, The University of Tokyo, Tokyo, Japan. [6] Research Initiative for Supra-Materials (RISM), Shinshu University, Nagano, Japan. ✉email: yanboli@uestc.edu.cn

Artificial photosynthesis that converts solar energy into sustainable energy and fuels is a highly desired solution to meet the increasing global energy demand and address the environmental issues related to the consumption of fossil fuels[1–7]. Semiconductor materials that absorb sunlight to generate electron-hole pairs are at the core of artificial photosynthetic devices[8–11]. Suppressing the bulk and interfacial nonradiative charge recombination in the semiconductor light absorbers is essential to achieve a high solar-to-fuel conversion efficiency[12–17]. Especially, the semiconductor/electrode and semiconductor/electrolyte interfaces play a decisive role in device performance because photogenerated charges must cross these interfaces to participate in the photosynthetic reactions. Most defects at the interfaces are high-dimensional deep-level defects, which usually have higher capture cross-sections than those of the point defects in the bulk of semiconductor[18]. Therefore, the nonradiative charge recombination at interfacial defect sites is detrimental to the device performance. In addition to interfacial defects, energy level alignment at the interfaces is also a key factor that affects the final device performance. Nonideal energy level alignment not only reduces device efficiency because of inefficient interfacial charge transfer, but it also deteriorates device stability due to interfacial charge accumulation which could promote self-oxidation/reduction of the semiconductor. Therefore, effective carrier management through interface engineering is essential to improve the efficiency and stability of photoelectrode for artificial photosynthesis[17–19].

Over the past decade, tremendous efforts have been devoted to interface engineering of various semiconductor photoelectrodes (e.g., Si[20,21], $\alpha$-Fe$_2$O$_3$[22], BiVO$_4$[23], Cu$_2$O[24]) to improve their efficiency and stability through the passivation of interfacial defects or formation of hetero-/homojunctions. Ideally, the semiconductor thin film light absorbers should be sandwiched by a n-type electron transport layer (ETL) and a p-type hole transport layer (HTL) to achieve efficient charge separation, similar to the "n-i-p" device architecture commonly used in thin-film photovoltaics[25]. However, this type of "n-i-p" device architecture is rarely adopted in photoelectrochemical (PEC) devices[26], especially for some of the emerging PEC materials. As one of the most promising photoanode materials for PEC water oxidation, Ta$_3$N$_5$ has a high theoretical solar-to-hydrogen (STH) efficiency of 15.9% owing to its wide absorption range (up to 600 nm) and proper band positions for water splitting. However, the full thermodynamic potential of Ta$_3$N$_5$ has yet to be unlocked. Although photocurrent approaching its theoretical value has been achieved[14], the onset potential (usually >0.5 V vs. RHE) is still far higher than its theoretical limit (<0 V vs. RHE). Despite that various strategies have been employed to improve the bulk charge transfer efficiency, the applied bias photon-to-current efficiency (ABPE) of Ta$_3$N$_5$-based photoanode is still below 3.31%[27,28]. For pristine Ta$_3$N$_5$ thin film based photoanode, the ABPE is still limited to 2.25%[29]. Promising results have been achieved by interface engineering of Ta$_3$N$_5$ photoanodes with TiO$_x$[14], AlO$_x$[30], and GaN[15,31,32] to enhance the photocurrent, lower the onset potential, and improve the long-term stability. However, these previous studies only focus on either the Ta$_3$N$_5$/electrode or the Ta$_3$N$_5$/electrolyte interfaces. It is still lacking a concerted effort to engineer both interfaces to form a "n-i-p" device architecture that mimics the sandwich structure in perovskite solar cells. Such a synergistic interface engineering strategy is expected to further improve the efficiency of Ta$_3$N$_5$ thin film based photoanode.

Herein, we propose to synergistically engineer the Ta$_3$N$_5$/electrode and Ta$_3$N$_5$/electrolyte interfaces with an n-type indium-doped gallium nitride (In:GaN) and a p-type magnesium-doped gallium nitride (Mg:GaN), respectively. The realization of the In:GaN/Ta$_3$N$_5$/Mg:GaN "n-i-p" heterostructure is possible because of the all-nitride compositions of the heterostructure that can be obtained by a single-step thermal nitridation process of their oxide precursors. The In:GaN and Mg:GaN interfacial layers not only facilitate selective charge extraction from Ta$_3$N$_5$ because of the desired energy band alignment of the formed "n-i-p" heterojunctions, they also passivate interfacial traps due to the formation of lattice-matched interfaces between Ta$_3$N$_5$ and the GaN layers. As a result, the ABPE is significantly improved from 2.29% for the pristine Ta$_3$N$_5$ photoanode to a record-high value of 3.46% for the In:GaN/Ta$_3$N$_5$/Mg:GaN photoanode, when both are modified with a highly active oxygen evolution reaction (OER) co-catalyst. Detailed mechanistic study reveals that the In:GaN layer mainly contributes to the enhanced bulk charge separation efficiency by selectively extracting photogenerated electrons from Ta$_3$N$_5$ through a mid-gap band generated by In doping in GaN. On the other hand, the Mg:GaN layer mainly contributes to improved surface charge injection efficiency by passivating surface traps in Ta$_3$N$_5$. These results demonstrate that interface engineering of semiconductor light absorbers with proper materials to construct band-aligned heterojunctions and passivate interface defects is an effective strategy to improve the solar-to-fuel conversion efficiency of artificial photosynthetic devices[33].

## Results and discussion

**Structural characterizations of In:GaN/Ta$_3$N$_5$/Mg:GaN heterostructure.** In:GaN/Ta$_3$N$_5$/Mg:GaN heterostructure films were fabricated by one-step thermal nitridation process of InO$_x$-GaO$_x$/TaO$_x$/Mg:GaO$_x$ thin films deposited by electron-beam (EB) evaporation and atomic layer deposition (ALD). Figure 1 shows the schematic diagram for the preparation of In:GaN/Ta$_3$N$_5$/Mg:GaN thin films. The detailed procedures are given in Methods. The multilayer structure of the InO$_x$-GaO$_x$/TaO$_x$/Mg:GaO$_x$ precursor film was resolved by using Auger electron spectroscopy (AES) in Supplementary Fig. 1. The scanning electron microscopy (SEM) images and X-ray diffraction (XRD) pattern of the prepared In:GaN/Ta$_3$N$_5$/Mg:GaN thin film on Nb substrate are shown in Supplementary Fig. 2. Polycrystalline film with grain size of several hundred nanometers and some small pores can been seen within the grains, which is characteristic of Ta$_3$N$_5$ thin film converted from TaO$_x$ precursor film due to volume contraction in the nitridation process[29]. The layered structure was not directly observed from the cross-sectional SEM image because of the thin thicknesses of the In:GaN (~5 nm) and Mg:GaN (~20 nm) layers and the lack of contrast between the three layers formed in a single-step nitridation process. Scanning transmission electron microscopy (STEM) was employed to further characterize the structure of the sample. Figure 2a shows the STEM energy dispersive x-ray spectroscopy (EDS) elemental mapping of the cross section of In:GaN/Ta$_3$N$_5$/Mg:GaN film deposited on Nb substrate and Fig. 2b shows the corresponding annular dark-field (ADF) STEM image. The STEM image and the EDS mapping of the metallic elements were overlapped in Fig. 2c. The thickness of the Ta$_3$N$_5$ layer is approximately 600 nm. A thin Ga-rich layer (~20 nm) on the top of the Ta$_3$N$_5$ layer was clearly revealed by the EDS mapping. However, the In:GaN layer at the bottom interface of the Ta$_3$N$_5$ layer was not clearly observed by EDS mapping. Therefore, AES cross sectional mapping and depth profile was employed to further characterize the sample (Supplementary Fig. 3). Similar to the EDS mapping results, AES mapping also only revealed the top Ga-rich layer, but not the bottom one. Nevertheless, AES depth profile clearly revealed the presence of a Ga-containing layer at the bottom interface of the Ta$_3$N$_5$ layer. It should be noted that Mg and In elements were also not detectable by either EDS or AES due to their low

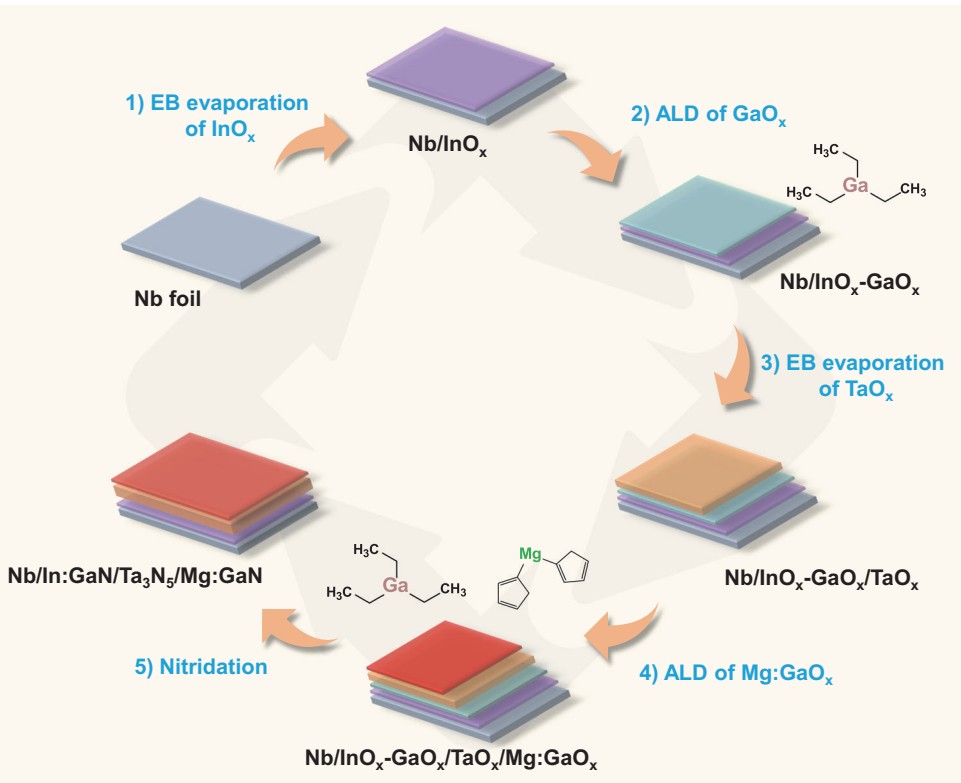

**Fig. 1 Schematic diagram for the preparation of In:GaN/Ta₃N₅/Mg:GaN heterostructure thin film. 1** EB evaporation of InOₓ thin layer on Nb substrate. **2** ALD of GaOₓ layer using TEG as precursor. **3** EB evaporation of TaOₓ layer. **4** ALD of Mg:GaOₓ layer using MgCp2 and TEG as precursors. **5** One-step thermal nitridation in NH₃ atmosphere at 1000 °C for 6 h.

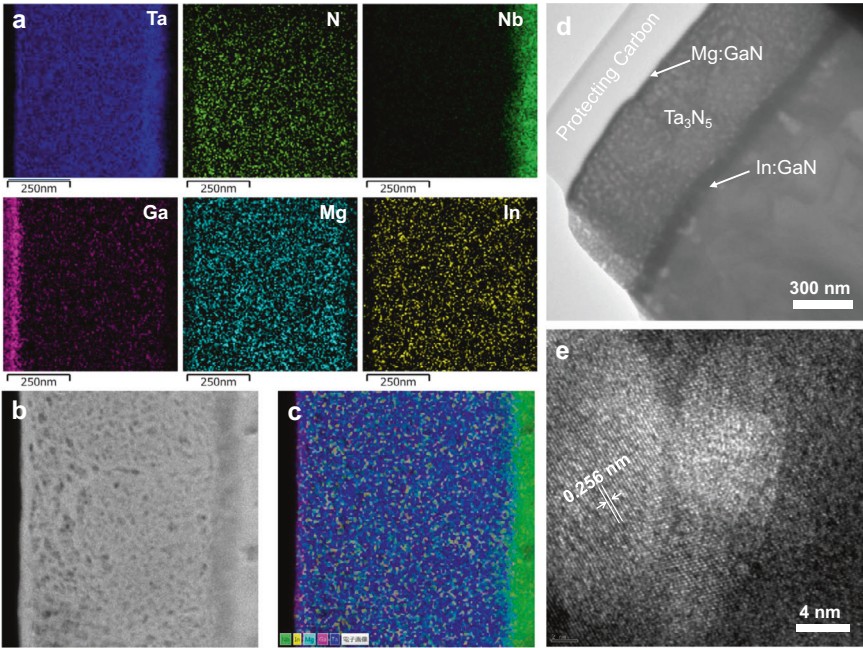

**Fig. 2 Structural properties of In:GaN/Ta₃N₅/Mg:GaN heterostructure thin film on Nb substrate. a** Cross-sectional STEM-EDS elemental mappings of Ta, N, Nb, Ga, Mg, and In. **b** Annular dark-field STEM image of the cross section of the sample. **c** Overlapping the AFD STEM image with the EDS mappings of the metallic elements. **d** Bright field TEM image of the film. **e** HRTEM image showing the lattice fringes of Ta₃N₅ with a spacing of 0.256 nm.

concentrations. The homogenous mapping results of Mg and In in Fig. 2a were likely resulted from noise spectra of the EDS instrument. Figure 2d shows the TEM image of the Nb/In:GaN/Ta₃N₅/Mg:GaN film, which exhibits intimate contact between each layers. The high-resolution TEM (HRTEM) image in Fig. 2e reveals lattice fringes with a spacing of 0.256 nm, corresponding to the (310) plane of Ta₃N₅. The lattice spacing is very close to that of GaN (002) plane (0.259 nm)[34], making it possible to form

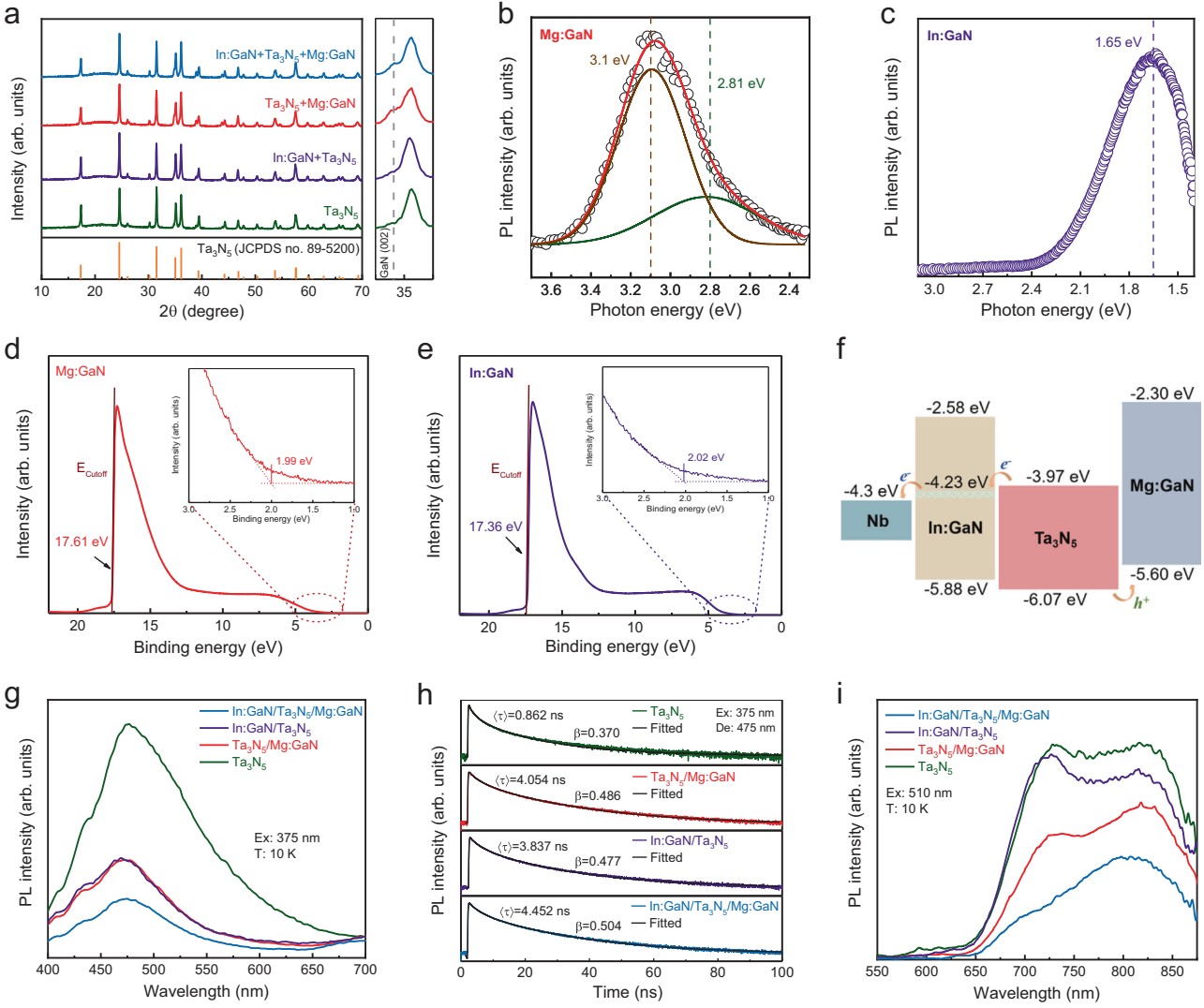

**Fig. 3 Spectroscopic characterizations of GaN and Ta$_3$N$_5$-based thin films. a** XRD patterns of different Ta$_3$N$_5$-based films deposited on quartz glass substrate. The narrow scan XRD patterns on the right side shows the (002) diffraction peaks of GaN. **b** PL spectra of Mg:GaN film deposited on quartz glass substrate under 270 nm LED excitation. and **c** PL spectra of In:GaN film deposited on quartz glass substrate under 375 nm laser excitation. **d** UPS spectrum of Mg:GaN deposited on Nb substrate. **e** UPS spectrum of In:GaN deposited on Nb substrate. **f** Schematic diagram of band structure for In:GaN/ Ta$_3$N$_5$/Mg:GaN film determined from UPS and UV–vis absorption measurements. **g** PL spectra and **h** TRPL spectra of four Ta$_3$N$_5$-based films on quartz glass substrate measured at 10 K under 375 nm laser excitation. **i** PL spectra of four Ta$_3$N$_5$-based films measured at 10 K under 510 nm laser excitation.

a lattice-matched interface between Ta$_3$N$_5$ and GaN, thus reducing the interface trap density, as demonstrated below.

**Spectroscopic characterizations**. XRD patterns of Ta$_3$N$_5$-based films with different layered structures deposited on quartz glass are exhibited in Fig. 3a. The main diffraction peaks can all be assigned to those of Ta$_3$N$_5$. No obvious change in the diffraction peaks is observed for films with different layered structures, indicating that the crystallinity of the Ta$_3$N$_5$ layer is well maintained after adding the GaN layers. However, a close-up of the peak at ~35° in the right panel of Fig. 3a reveals a distinct shoulder peak at ~34.7°, corresponding to the diffraction peak of GaN (002), when Mg:GaN layer is formed on top of the Ta$_3$N$_5$ film[15]. This indicates that these GaN-based layers have been successfully synthesized in a single-step nitridation process. To further confirm that the GaN layers can be formed by nitridation of GaO$_x$ precursor films, In:GaN and Mg:GaN thin films were deposited directly on quartz glass substrates by similar processes.

The XRD patterns of the In:GaN and Mg:GaN films in Supplementary Fig. 4 match that of hexagonal GaN (JPCDS no. 50-0792), while a weak peak belonging to GaInO$_3$ impurity phase at ~35.1° is found in the In:GaN film. A slight peak shift to lower angle is observed the In:GaN film, indicating In is incorporated into GaN lattice because the ionic radius of In$^{3+}$ is large than that of Ga$^{3+}$. The presence of Mg and In in the synthesized GaN films is further confirmed by X-ray photoelectron spectroscopy (XPS) results (Supplementary Figs. 5 and 6). Photoluminescence (PL) spectroscopy was employed to investigate the fluorescence properties of Mg:GaN and In:GaN films (Fig. 3b, c). The PL spectrum of Mg:GaN film can be deconvolved into two sub-peaks centered at ~3.1 and ~2.8 eV, which are typical fluorescence emission peaks related to the isolated Mg$_{Ga}$ centers and Mg-related deep levels, respectively[35]. In contrast, the PL spectrum of In:GaN film shows a emission peak at ~1.65 eV, which suggests In-doping induces mid-gap states in GaN. The In-induced mid-gap states could be beneficial to electron transport from the

Ta$_3$N$_5$ light absorber through the In:GaN layer to the Nb electrode, as shown below.

The band positions of the Mg:GaN and In:GaN films were determined using ultraviolet photoelectron spectroscopy (UPS) (Fig. 3d, e). By subtracting the cut-off energies ($E_{Cutoff}$) of the secondary electrons from the He I excitation energy (21.22 eV), the Fermi levels of Mg:GaN and In:GaN were obtained at 3.61 and 3.86 eV below the vacuum level, respectively. The low-binding energy edges of the UPS spectra reveal that the valence bands of Mg:GaN and In:GaN are 1.99 and 2.02 eV below their Fermi levels, respectively. The Tauc plots of the absorption spectra revealed that the optical bandgaps of the In:GaN and Mg:GaN films are both ~3.30 eV (Supplementary Fig. 7). Combining these data with the band positions of pristine Ta$_3$N$_5$ obtained in our previous study[27], the band alignment of the Nb/In:GaN/Ta$_3$N$_5$/Mg:GaN layered structure is obtained and shown in Fig. 3f. The band alignment shows that pristine GaN is not suitable for electron extraction from Ta$_3$N$_5$ because its conduction band lies too high above that of Ta$_3$N$_5$. However, In-doping induces a mid-gap band at around −4.23 eV in In:GaN, which could serve as a conductive pathway for electrons. The ability of the In-induced inter-gap state to act as a channel for electrons was further verified by measuring cyclic voltammetry of a compact In:GaN film in the presence of a scavenger (Supplementary Fig. 8). Such a conduction mechanism through inter-gap defect states has previously been demonstrated with amorphous TiO$_2$ coated on Si, GaAs, and GaP photoanodes[1]. Consequently, the photon-generated electrons and holes in the Ta$_3$N$_5$ layer can be efficiently extracted through the In:GaN and Mg:GaN layers, respectively.

To reveal the effect of the In:GaN underlayer and Mg:GaN overlayer on the carrier dynamics of Ta$_3$N$_5$, low-temperature and time-resolved PL (TRPL) spectroscopies were employed to study samples with different layered structures. Due to the strong optical anisotropy of its orthorhombic lattice, Ta$_3$N$_5$ shows two intrinsic bandgaps of ~2.1 eV along the $a$-axis and ~2.6 eV along the $b$- and $c$-axes[29]. We found that at low temperature (10 K) under 375-nm laser excitation the PL of Ta$_3$N$_5$ was dominated by an emission peak at ~475 nm (~2.6 eV), corresponding to the near band edge radiative recombination along the $b$- and $c$-axes. Therefore, the intensities and lifetimes of this peak were measured to probe the carrier dynamics in Ta$_3$N$_5$-based samples with different layered structures. The steady-state PL spectra in Fig. 3g show that the PL intensity is quenched when Ta$_3$N$_5$ film is modified with either In:GaN or Mg:GaN layers, suggesting their ability to extract photocarriers from Ta$_3$N$_5$. Moreover, the TRPL results reveal that the In:GaN and Mg:GaN layers also play a role in defect passivation besides carrier extraction. All the TRPL decay curves in Fig. 3h are well-fitted with a stretched-exponential decay (Supplementary Fig. 9). This decay law is typically observed in disordered systems and it is believed to be due to the dispersion diffusion of photoexcited carriers[36]. The degree of disorder depends on the trap density and is measured by the stretching component β (0 < β < 1). The higher the β value the less disorder (lower trap density) the semiconductor[37,38]. The details about the stretched-exponential decay fitting are given in Supplementary Note 1. The fitted results in Supplementary Table 1 show that both the average lifetime <τ> and the stretching component β increase significantly when the Ta$_3$N$_5$ film is modified with In:GaN and Mg:GaN layers. This indicates the trap density of the Ta$_3$N$_5$ layer is lowered by the interface and surface modification with the In:GaN and Mg:GaN layers. To further clarify the change of trap density in these samples, defect-related PL spectra were measured under excitation of a 510-nm laser at 10 K (Fig. 3i). All the PL spectra could be deconvolved into two distinctive emission peaks centered at around 720 and

820 nm (Supplementary Fig. 10), which are ascribed to defect emissions of nitrogen vacancies and reduced Ta$^{3+}$ defects, respectively, according to our previous study[29]. The intensity of the defect-related PL emission significantly decreases with In:GaN and Mg:GaN modification, which indicates the trap density in the Ta$_3$N$_5$ layer is reduced. In contrast to the abrupt lattice termination on the surface of bare Ta$_3$N$_5$ layer, the In:GaN and Mg:GaN modification resulted in lattice-matched GaN/Ta$_3$N$_5$ interfaces, thus reducing the defect density near the interfaces[15].

**PEC performance of In:GaN/Ta$_3$N$_5$/Mg:GaN photoanode.** The PEC performance of Ta$_3$N$_5$ films with or without interface engineering with In:GaN and Mg:GaN layers was tested after modifying their surfaces with a borate-intercalated nickel cobalt iron oxyhydroxide (NiCoFe-B$_i$) OER co-catalyst. The co-catalyst modification is essential to improve the activity and stability of the photoanodes (Supplementary Fig. 11). Figure 4a shows the photocurrent-potential ($J$–$V$) curves for Ta$_3$N$_5$-based photoanodes with different layer structures. The detailed parameters of the $J$–$V$ curves were listed in Supplementary Table 2. The Ta$_3$N$_5$ photoanode exhibited a relatively low photocurrent density of 7.5 mA cm$^{-2}$ at 1.23 V vs. RHE. With either In:GaN or Mg:GaN modification, the photocurrent densities of the photoanodes were increased to over 8 mA cm$^{-2}$ at 1.23 V vs. RHE and the fill factors of the $J$–$V$ curves were improved obviously. The most significant enhancement in photocurrent density was achieved when both the bottom and top interfaces of the Ta$_3$N$_5$ film were modified with In:GaN and Mg:GaN, yielding a value of 9.3 mA cm$^{-2}$ at 1.23 V vs. RHE. The enhancement was ascribed to the effective charge separation and interface defect passivation effect of the In:GaN and Mg:GaN layers. The photocurrent onset potential, measured at a steady-state photocurrent density of ~20 μA cm$^{-2}$ (Fig. 4b), was also found to shift cathodically from 0.47 V vs. RHE for the Ta$_3$N$_5$ photoanode to 0.38 V vs. RHE for the In:GaN/Ta$_3$N$_5$/Mg:GaN photoanode. Consequently, a maximum ABPE of 3.46% was achieved for the In:GaN/Ta$_3$N$_5$/Mg:GaN photoanode (Fig. 4c), which is the highest ABPE reported so far for Ta$_3$N$_5$-based photoanode, to the best of our knowledge (Supplementary Fig. 12). The In:GaN/Ta$_3$N$_5$/Mg:GaN photoanodes samples also showed good reproducibility in PEC performance: the average ABPE for a batch of eight samples is 3.31% with a standard deviation of 0.11% (Supplementary Fig. 13).

The stability of NiCoFe-B$_i$ modified Ta$_3$N$_5$ and In:GaN/Ta$_3$N$_5$/Mg:GaN photoanodes was tested at 1.0 V vs. RHE in 1 M KOH under simulated sunlight. The photocurrent of the Ta$_3$N$_5$ photoanode decreased continuously due to the self-oxidation of the Ta$_3$N$_5$ surface (Supplementary Fig. 14). In contrast, the In:GaN/Ta$_3$N$_5$/Mg:GaN photoanode generated a stable photocurrent at ~8.9 mA cm$^{-2}$ for 160 min (Fig. 4d). The improved stability is ascribed to the passivation effect of the Mg:GaN layer, which prevents the self-oxidation of the Ta$_3$N$_5$ surface, a key factor of performance degradation in Ta$_3$N$_5$ photoanode[19]. For longer-term stability test, the In:GaN/Ta$_3$N$_5$/Mg:GaN photoanode maintained 80% of its initial photocurrent after 10 h and 70% after 15 h (Supplementary Fig. 15). Figure 4e plots the wavelength dependence of the incident photon-to-current conversion efficiency (IPCE) of the In:GaN/Ta$_3$N$_5$/Mg:GaN photoanode. The IPCE values were between 70% and 90% in a broad spectrum range of 400–550 nm, which indicates the impressive photon conversion efficiency of In:GaN/Ta$_3$N$_5$/Mg:GaN photoanode in the visible range. An integrated photocurrent density of ~9.0 mA cm$^{-2}$ was obtained by integrating the IPCE values over the standard AM 1.5 G solar spectrum, which matches well with the measured photocurrent density in Fig. 4a (~8.9 mA cm$^{-2}$). Figure 4f shows the amount of O$_2$ evolved from the In:GaN/

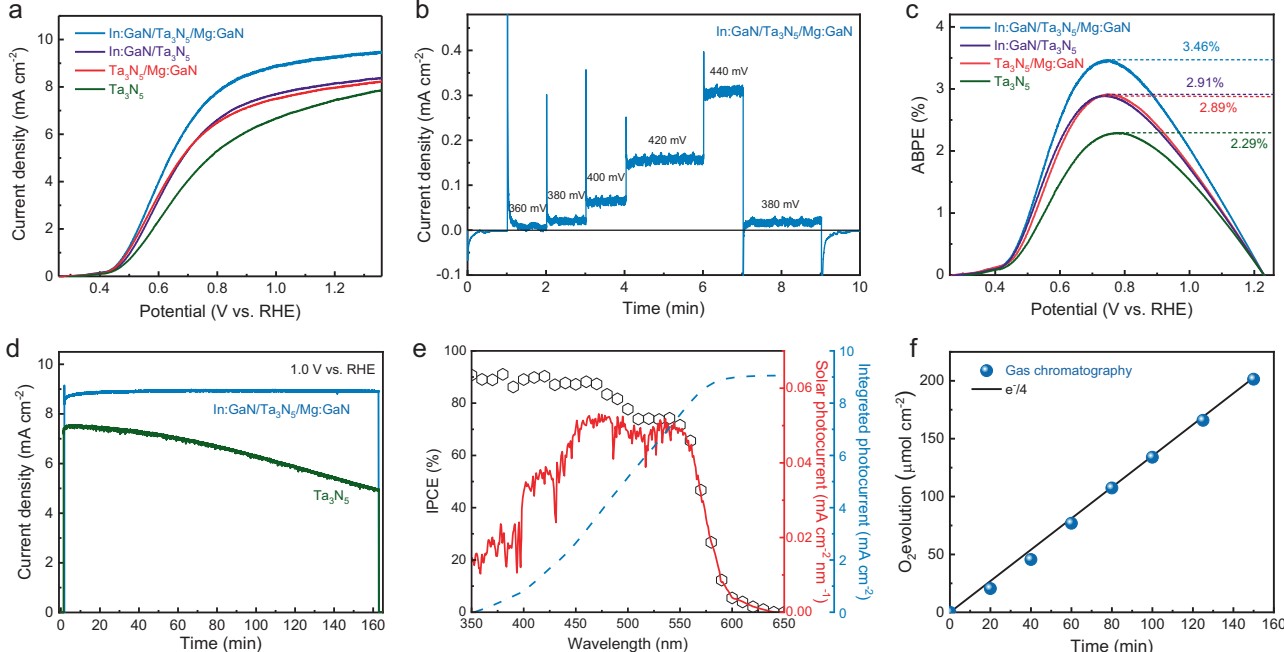

**Fig. 4 PEC performance of the In:GaN/Ta₃N₅/Mg:GaN photoanode on Nb substrate. a** $J$–$V$ curves of Ta₃N₅-based photoanodes with different layered structures. All the photoanodes were modified with NiCoFe-B$_i$ co-catalyst and tested in 1 M KOH electrolyte under AM 1.5 G illumination. **b** The steady-state photocurrent of In:GaN/Ta₃N₅/Mg:GaN photoanode under low-bias conditions. **c** ABPE curves calculated from the J-V curves in **a**. **d** Stability of the pristine Ta₃N₅ and In:GaN/Ta₃N₅/Mg:GaN photoanodes measured at an applied potential of 1.0 V vs. RHE. **e** IPCE spectrum of the In:GaN/Ta₃N₅/Mg:GaN photoanode at 1.0 V vs. RHE and the corresponding solar photocurrent and integrated photocurrent calculated using the standard AM 1.5 G solar spectrum (ASTM G173-03). **f** Amount of O₂ evolved from the In:GaN/Ta₃N₅/Mg:GaN photoanode under an applied potential of 1.0 V vs. RHE.

Ta₃N₅/Mg:GaN photoanode quantified by the gas chromatography (GC). The dark line plots a quarter of the electron numbers ($e^-$/4) calculated from the measured photocurrent (Supplementary Fig. 16), which gives the expected amount of O₂ evolution assuming 100% Faraday efficiency. The GC data points align well with the calculated $e^-$/4 line, indicating nearly unity Faraday efficiency was achieved with the In:GaN/Ta₃N₅/Mg:GaN photoanode. This result demonstrates that the measured photocurrent is indeed contributed by OER and the ABPE calculated from $J$–$V$ curve does represent the STH conversion efficiency of the In:GaN/Ta₃N₅/Mg:GaN photoanode.

To further clarify the roles of In and Mg doping in the GaN layers in the heterojunction structure, pristine GaN layers were employed for interface engineering for comparison. The GaN/Ta₃N₅ photoanode showed a degraded PEC performance compared with the Ta₃N₅ photoanode (Supplementary Fig. 17a), which is likely due to the mismatched band alignment that formed a large potential barrier on the back of the photoanode. In contrast, the In:GaN/Ta₃N₅ photoanode showed obviously improved PEC performance compared with the Ta₃N₅ photoanode, which is ascribed to the generation of a conductive mid-gap band in GaN by In doping, as revealed in Fig. 2f. The Ta₃N₅/GaN photoanode showed a slightly enhanced PEC performance compared with Ta₃N₅ photoanode (Supplementary Fig. 17b), consistent with the previous report[15]. Nevertheless, the improvement was not as great as that achieved with Ta₃N₅/Mg:GaN photoanode. These results demonstrate that proper doping of the interfacial GaN layers is crucial to improve the carrier separation efficiency and hole injection efficiency of Ta₃N₅ photoanode.

**Electrochemical characterizations of the heterostructure.** Surface injection efficiency ($\eta_{inj}$) and bulk charge separation efficiency ($\eta_{bulk}$) were determined to further investigate the effect of In:GaN and Mg:GaN layers. The details about the calculation of

$\eta_{inj}$ and $\eta_{bulk}$ are given in Supplementary Note 2. $\eta_{bulk}$ represents the fraction of photo-generated holes that arrive at the photoanode/electrolyte without recombination in the bulk, while $\eta_{inj}$ denotes the fraction of those holes that successfully inject into the electrolyte for water oxidation. The $J$–$V$ curves of four different photoanodes were measured in 1 M KOH electrolyte with or without 0.5 M H₂O₂ as a hole scavenger (Supplementary Fig. 18). The absorption photocurrent density ($J_{abs}$) of each photoanode was calculated based on the UV–visible absorption spectrum by assuming 100% absorbed photon-to-current conversion efficiency (Supplementary Fig. 19). The potential dependent $\eta_{bulk}$ curves in Fig. 5a reveal that the improvement of $\eta_{bulk}$ is mainly attributed to the modification of In:GaN at the bottom interface of Ta₃N₅ film. This is reasonable as In:GaN selectively extracts photo-generated electrons from the Ta₃N₅ film, thus improving the bulk electron-hole separation efficiency. Meanwhile, the potential dependent $\eta_{inj}$ curves in Fig. 5b reveal that the improvement of $\eta_{inj}$ is mainly originated from the modification of Mg:GaN on the surface of Ta₃N₅ film. These results have helped us distinguish the respective role of In:GaN and Mg:GaN in enhancing the PEC performance of the heterostructure photoanode.

Photoelectrochemical impedance spectroscopy (PEIS) was employed to compare the charge transfer processes between Ta₃N₅ and In:GaN/Ta₃N₅/Mg:GaN photoanodes[14,22]. The PEIS Nyquist plots in Fig. 5c show that the radius of the semicircle curve for In:GaN/Ta₃N₅/Mg:GaN photoanode dramatically reduces compared with that of Ta₃N₅ photoanode, which indicates the improvement of charge separation efficiency and injection efficiency. A typical two-RC-unit equivalent circuit model was used to fit the Nyquist plot of Ta₃N₅ photoanode, while a more complex model with the addition of a heterojunction RC-unit was used to fit the Nyquist plot of In:GaN/Ta₃N₅/Mg:GaN photoanode (Supplementary Fig. 20)[22]. The PEIS Nyquist plots were well-fitted using the equivalent circuit models

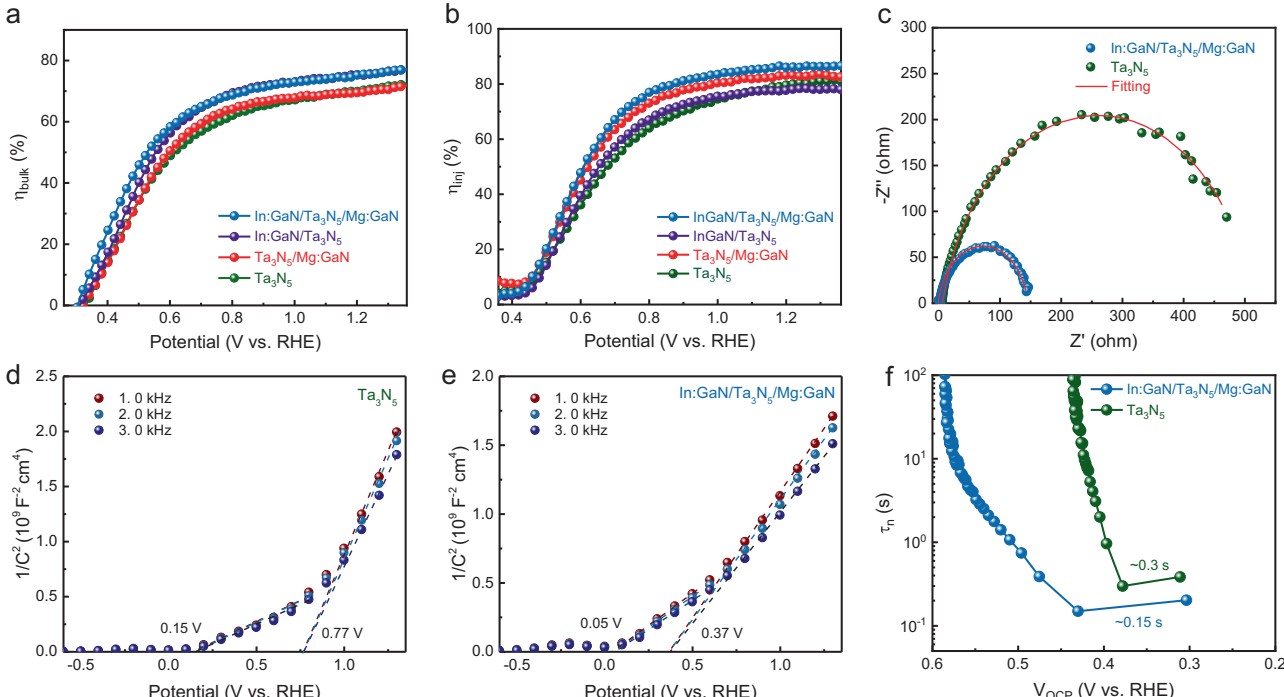

**Fig. 5 Electrochemical characterizations of the Ta₃N₅-based thin films with different layered structures on Nb substrate. a** Bulk charge separation efficiency ($\eta_{bulk}$). **b** Surface injection efficiency ($\eta_{inj}$). **c** PEIS of Ta₃N₅ and In:GaN/Ta₃N₅/Mg:GaN photoanodes measured in 1 M KOH electrolyte at 1.0 V vs. RHE under AM 1.5 G illumination. Red lines show the fitting of the PEIS data. M-S plots of **d**, Ta₃N₅ photoanode, and **e** In:GaN/Ta₃N₅/Mg:GaN photoanode. The M-S plots were measured in the dark without co-catalyst. **f** Carrier lifetimes derived from OCP-decay curves at the light on-off transient for Ta₃N₅ and In:GaN/Ta₃N₅/Mg:GaN photoanodes.

and the detailed fitting parameters are listed in Supplementary Table 3. The trapping resistance ($R_{trap}$) and the charge transfer resistance ($R_{ct}$) of the In:GaN/Ta₃N₅/Mg:GaN photoanode are 6.0 and 23.0 Ω, respectively. In contrast, the Ta₃N₅ photoanode shows a significantly larger $R_{trap}$ and $R_{ct}$ values of 449.3 and 503.7 Ω, respectively. Although additional resistance ($R_{hete}$) is generated in the In:GaN/Ta₃N₅/Mg:GaN photoanode due to the hetero-junctions, it is relatively small (1.13 Ω). These results demonstrate that the separation and injection efficiencies are indeed improved by interface engineering with the In:GaN and Mg:GaN layers.

Mott-Schottky (M-S) analysis was employed to probe the change of surface energetics of the photoanodes with and without interface engineering. The M-S plot of Ta₃N₅ photoanode in Fig. 5d shows that the flat-band potential is at 0.15 V vs RHE. However, the slope of the M-S plot changes at higher applied potential region (>0.77 V vs RHE), suggesting that the band bending develops to a different extent in different potential regions (Supplementary Fig. 21a). At low potential region (<0.77 V vs RHE), the Fermi level pinning due to the existence a distribution of surface states causes the band bending to develop in a lesser extent while part of the applied potential drops at the Helmholtz layer[39]. At high potential region (>0.77 V vs RHE), the decreased density of surface states allows for the band bending to develop in a higher extent, leading to a larger slope in the M-S plot. For the In:GaN/Ta₃N₅/Mg:GaN photoanode in Fig. 5e, the flat-band potential shifts cathodically to 0.05 V vs RHE while the transition of the two slopes happens at 0.37 V vs RHE. The lower applied potential to overcome the Fermi level pinning suggests there is a lower density of surface states due to the passivation effect of the Mg:GaN layer. The band bending can be developed in a larger extend at lower applied potential region, thus reducing the bias-potential requirement for the water oxidation process (Supplementary Fig. 21b). The open circuit potential (OCP)

decay profile was employed to further demonstrate the passivation of interfacial traps by interface engineering[40–42]. Supplementary Fig. 22 shows the OCP decay profiles of Ta₃N₅ and In:GaN/Ta₃N₅/Mg:GaN photoanodes. Compared with Ta₃N₅ photoanode, the amplitude of the OCP decay is significantly larger in In:GaN/Ta₃N₅/Mg:GaN photoanode, which indicates larger band bending at photoanode/electrolyte interface under illumination[41]. The carrier lifetimes are derived from the OCP decay curves (Supplementary Note 3) and plotted against OCP in Fig. 5f. At the transient when illumination was switched off, the carrier lifetime in In:GaN/Ta₃N₅/Mg:GaN photoanode was reduced by a factor of two compared to that in Ta₃N₅ photoanode. The faster decay kinetics upon switching off the illumination suggests the reduced charge trapping at the interfaces in In:GaN/Ta₃N₅/Mg:GaN photoanode. The larger band bending and reduced charge trapping in the depletion region near photoanode/electrolyte interface accelerate charge separation, leading to an improved PEC activity of the In:GaN/Ta₃N₅/Mg:GaN photoanode.

In summary, an "n-i-p" heterostructure In:GaN/Ta₃N₅/Mg:GaN photoanode was prepared by one-step thermal nitridation process of InOₓ-GaOₓ/TaOₓ/Mg:GaOₓ precursor film deposited by EB evaporation and ALD. Mechanistic studies revealed that the n-type In:GaN layer mainly improved the bulk charge separation efficiency by selectively extracting photogenerated electrons from Ta₃N₅ through a conductive mid-gap band induced by In-doping. Meanwhile, the p-type Mg:GaN layer mainly contributed to the enhanced surface charge injection efficiency of the photoanode by passivating the traps on the surface of Ta₃N₅. Benefiting from the improved bulk charge separation efficiency and surface injection efficiency, the In:GaN/Ta₃N₅/Mg:GaN photoanode yielded a maximum ABPE of 3.46%, which was significantly improved from 2.29% of the Ta₃N₅

photoanode. Owning to the passivation effect of Mg:GaN layer, the stability of the In:GaN/Ta$_3$N$_5$/Mg:GaN photoanode was also drastically improved. The application of In:GaN and Mg:GaN as interfacial modification layers could be extended to other oxy-nitride semiconductors (e.g., TaON, BaTaO$_2$N, LaTiO$_2$N) to further improve their conversion efficiency for PEC water splitting. Finally, it is proposed that effective carrier management through interface engineering should be generally considered to improve the efficiency and stability of photoelectrode for artificial photosynthesis.

## Methods

**Synthesis of In:GaN/Ta$_3$N$_5$/Mg:GaN thin films**. Indium oxide (InO$_x$) layer with a nominal thickness of 2 nm was first deposited on Nb foils (10 × 10 × 0.1 mm$^3$), quartz glass (10 × 10 × 1 mm$^3$), or Si substrates (10 × 10 × 0.525 mm$^3$) by EB eva-poration (Angstrom Engineering AMOD) at a deposition rate of 0.5 Å/s using In$_2$O$_3$ (99.99% in purity) as the source material. Then a gallium oxide (GaO$_x$) layer with a thickness of ~5 nm was deposited on top of the InO$_x$ layer by plasma-enhanced ALD (Picosun R-200 Advanced) using triethyl gallium (TEG, 99.99%) as the source and O$_2$ plasma as the oxidizer. The TEG source temperature was maintained at 25 °C and the substrate temperature was 250 °C. A typical ALD sequence consisted of TEG exposure (0.5 s)/N$_2$ purge (5 s)/O$_2$ plasma exposure (12 s)/N$_2$ purge (5 s). Subsequently, a tantalum oxide (TaO$_x$) layer with a thickness of 700 nm was deposited on top of the GaO$_x$ layer by EB evaporation at a deposition rate of 4 Å/s using Ta$_2$O$_5$ (99.99%) as the source material. During deposition, O$_2$ gas (99.999%) was introduced into the deposition chamber at a flow rate of 5 sccm and the working pressure was approximately 2 × 10$^{-4}$ Torr. After-wards, the InO$_x$-GaO$_x$/TaO$_x$ precursor films were dipped into a H$_2$O$_2$ aqueous solution (30%) for 2 h to oxidize the slightly reduced TaO$_x$ films. Subsequently, a magnesium-doped GaO$_x$ (Mg:GaO$_x$) layer with a thickness of ~20 nm was deposited on top of the TaO$_x$ layer by plasma-enhanced ALD using TEG and bis(cyclopentadienyl)magnesium (MgCp2, 99.99%) as the sources. The MgCp2 source temperature was maintained at 80 °C and the substrate temperature was 250 °C. An ALD "super-cycle" consisted of: 5 cycles of TEG exposure (0.5 s)/N$_2$ purge (5 s)/O$_2$ plasma exposure (12 s)/N$_2$ purge (5 s), followed by one cycle of MgCp2 exposure (1.6 s)/N$_2$ purge (5 s)/O$_2$ plasma exposure (12 s)/N$_2$ purge (5 s). The super-cycle was repeated 58 times to reach the target thickness. Finally, In:GaN/Ta$_3$N$_5$/Mg:GaN thin films were obtained by thermal nitridation of the InO$_x$-GaO$_x$/TaO$_x$/Mg:GaO$_x$ thin films in a horizontal quartz tube (inner diameter: 21 mm) furnace (MTI OTF-1200X). The quartz tube was purged with 100 sccm NH$_3$ (99.999%) flow at ambient pressure. The temperature was ramped at a rate of 10 °C min$^{-1}$ from room temperature to 1000 °C, maintained for 6 h, and cooled down naturally to room temperature.

**Structural and spectroscopic characterizations**. XRD was measured with a Thermo Scientific ARL™ EQUINOX 1000 in θ–2θ configuration using a Cu Kα radiation source operated at 40 kV and 30 mA. SEM images were taken with a ZEISS Crossbeam 340. UV–vis spectra were measured with a SHIMADZU UV-1900. XPS was performed by PHI 5000 VersaProbe III with a monochromatic Al Kα X-ray source with the beam size of 200 μm. Charge compensation was achieved by the dual beam charge neutralization and the binding energy was corrected by setting the binding energy of the hydrocarbon C 1 s feature to 284.8 eV. UPS was performed by PHI 5000 VersaProbe III with He I source (21.22 eV) under an applied negative bias of 9.0 V. Cross sectional samples were prepared by focused ion beam (JEOL JIB-4600F) etching and followed by 0.5–1.5 kV gentle Ar milling with a low angle ion milling and polishing system (JEOL 1010) for finishing. The cross-sectional STEM and HRTEM images and EDS mapping were taken with a JEOL JEM-2800 equipped with X-MAX 100TLE SDD detector (Oxford Instruments). AES was performed by PHI 710 Scanning Auger Nanoprobe, equipped with a thermally assisted Schottky field-emission electron gun and a coaxial cylindrical mirror analyzer. AES depth profiling was accomplished using the mono Ar ion source operated at 1 kV and rostered over a 2 × 2 mm area. PL and TRPL spectra were measured by Picoquant FluoTime 300 using 270 nm LED or 375/510 nm picosecond pulsed lasers as the excitation sources. The temperature of the samples was cooled to ~10 K using a closed-cycle He cryostat (ARS DE-202).

**Fabrication of In:GaN/Ta$_3$N$_5$/Mg:GaN photoanodes**. The In:GaN/Ta$_3$N$_5$/Mg:GaN thin films deposited on Nb substrates were used for the fabrication of photoanodes. Electrical contact was established by soldering a copper wire to the back side of the Nb substrate using indium. The edges and back side of the samples were then encapsulated by a rapid-curing epoxy (Araldite). The active area of the photoanode was approximately 0.94 cm$^2$. After completely curing for at least 12 h, NiCoFe-B$_i$ OER co-catalyst was deposited on the surface of the photoanode by a photo-assisted electrochemical deposition method. Borate buffer solution (pH ~10) prepared by mixing 0.6 M KOH and 1 M H$_3$BO$_3$ was used as the electrolyte. After purging the electrolyte with Ar for 15 min, 0.5 mM Co(NO$_3$)$_2$·6H$_2$O, 2 mM NiSO$_4$·6H$_2$O and 0.8 mM FeSO$_4$·7H$_2$O were sequentially added to the electrolyte.

The deposition process was carried out at a constant current density of 30 μA cm$^{-2}$ for 10 min under AM 1.5 G illumination (SAN-EI Electric, XES-40S3-TT). After deposition, the samples were rinsed with deionized water.

**PEC measurement**. All the PEC measurements were conducted using a potentiostat (Bio-Logic SP-200) in three-electrode configuration with a Pt counter electrode and a Hg/HgO reference electrode. All measured potentials versus Hg/HgO were converted to RHE scale according to the Nernst equation. The Pt cathode chamber was separated from the photoanode chamber using a Nafion 117 membrane. Simulated sunlight from a solar simulator (SAN-EI ELECTRIC, XES-40S3-TT) with intensity calibrated to 100 mW cm$^{-2}$ (AM1.5 G) using a certified reference cell (Konica-Minolta AK-200) was used as the light source. J-V curves were recorded under a cathodic scan at a rate of 10 mV s$^{-1}$ at 283 K in 1 M KOH (pH 13.6) solution under magnetic stirring and Ar bubbling. ABPE was calculated from the J-V curves with the equation: ABPE = [(1.23 − V$_{app}$) × J$_{light}$/P$_{light}$] × 100%, where V$_{app}$ is the applied potential (versus RHE), J$_{light}$ is the photocurrent density under AM 1.5 G light, P$_{light}$ equals to 100 mW cm$^{-2}$ for the simulated sunlight. IPCE spectrum was measured at 1 V vs RHE from 350 to 650 nm with an interval of 10 nm using a monochromatic light source (Zolix Sirius 300 P). The IPCE value at each wavelength (λ) was calculated by: IPCE = [(1240/λ) × (J$_{light}$ − J$_{dark}$)/P$_{light}$] × 100%, where J$_{light}$ was the measured photocurrent density, J$_{dark}$ was the current density in the dark, P$_{light}$ was the irra-diance of the monochromatic light. The amount of oxygen evolved from the photoanode was quantified by a gas chromatography (Shimadzu GC-2014). PEIS was performed at 1.0 V vs RHE under AM 1.5 G simulated sunlight in frequency range of 1 Hz to 1 MHz. Fitting of the Nyquist plots was performed by ZSimDemo software. M-S plots were measured in the potential range of −0.6–1.3 V vs RHE with an AC amplitude of 10 mV in the frequency range of 1.0–3.0 kHz under dark conditions. The OCP decays at the light on/off transient were recorded after illu-minating the sample for 10 min in 1 M KOH.

## Data availability

Source data are provided with this paper.

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

## Acknowledgements

This work was supported by the National Natural Science Foundation of China (No. 21872019). M.N., N.S., and K.D. acknowledge the Artificial Photosynthesis Project (ARPChem) of the New Energy and Industrial Technology Development Organization (NEDO). A Part of this study was supported by the University of Tokyo Advanced Characterization Nanotechnology Platform in the Nanotechnology Platform Project sponsored By the Ministry of Education, Culture, Sports, Science and Technology (MEXT), Japan (JPMXP09-A-20-UT-0004).

## Author contributions

Y.L. and J.F. proposed the project; Y.L., N.S., and K.D. supervised the project; J.F. carried out the device fabrication and PEC tests with the assistance of Z.F., N.P., Y.X., and C.F.; M.N performed the STEM and HRTEM experiments; H.J. performed the AES experiment. Y.L. and J.F. analyzed the results and wrote the manuscript. All authors commented on and revised the manuscript.

## Competing interests

The authors declare no competing interests.
