## [Peer Review File · Nature Communications]

Interface engineering of Ta₃N₅ thin film photoanode for highly efficient photoelectrochemical water splittingREVIEWER COMMENTS

Reviewer #1 (Remarks to the Author):

This work by Li et al demonstrates a record-high applied bias photon-to-current efficiency of 3.46% for photoelectrochemical water splitting over Ta₃N₅-based photoanode. A n-i-p type heterostructure is formed by modifying the Ta₃N₅ bottom and top interfaces with n-type In-doped GaN and p-type Mg-GaN layers. The result is very exciting, especially the demonstration of effective electron and hole transport layers that potentially can be applied to other (oxy-)nitride semiconducting light absorbers. Overall, I find the paper to be well written and the characterization work also well carried out according to standard procedures. The effect of interface engineering is well worth publishing in Nature Communications. There are only a few questions the authors should clarify before acceptance:

- 1) In a previous work by the authors (Nature Catalysis, 3, 932-940, 2020), a Mg-doped Ta₃N₅ thin film was used as the light absorber to achieve an ABPE of 3.31%. The Mg-doped Ta₃N₅ seems to be more efficient than the pristine Ta₃N₅ used in this study. Why not using this type of Mg-doped Ta₃N₅ as the light absorber to construct the n-i-p heterostructure?
- 2) For the synthesis of In-doped GaN layer a InO_x-GaO_x bilayer was deposited by electron beam evaporation and ALD respectively as the precursor, while for the synthesis of Mg-doped GaN layer both MgO_x and GaO_x were deposited in a single ALD process. The latter seems to be better in achieving precursor films with more uniform doping. Why not using ALD for the deposition of InO_x precursor as well?
- 3) The OER cocatalyst also plays a role in passivating surface defects in many photoanode. To better demonstrate the effectiveness of the surface passivation effect of the Mg:GaN layer and to show the effect of the cocatalyst, it is necessary to test the PEC performance of the photoanodes without cocatalyst modification.

Reviewer #2 (Remarks to the Author):

The manuscript "Interface Engineering of Ta₃N₅ thin film photoanode for highly efficient photoelectrochemical water splitting" describes that the simultaneous modification of Ta₃N₅ photoanodes with the deposition of Mg:GaN in the electrolyte interface and In:GaN in the back contact leads to a benchmark performance for solar water oxidation. The authors display good care when it comes to characterize the materials and interpret the results. However there some open questions that

must be addressed in a major revision before further consideration of this paper for publication in Nature Communications.

- 1) In the introduction the authors motivate the importance of interface engineering invoking studies on perovskite solar cells. This is quite surprising given the large body of work on that particular area in the field of photoelectrochemical water splitting. I would encourage the authors to motivate the need for such “back” and “surface” engineering citing studies from the field of photoelectrochemistry since these align better with the topic of this study.
- 2) In page 4, the authors mentioned that “the full thermodynamic potential of Ta₃N₅ has yet to be unlocked”. I would recommend rephrasing this statement or dedicate some lines to explain what is considered “full thermodynamic potential”, perhaps the authors refer to the maximum photovoltage that could be delivered or the photocurrent, in such a case, it could be clearer to state those values to compare with the current state of the art.
- 3) Although the authors show cross sectional figures, I could not find the thickness of the Mg:GaN, In:GaN or even Ta₃N₅. The authors need to include these values to better interpret and understand the results, as I will mention below.
- 4) Correct in page 9 “could serve as a conductive pathway” by “could serve as a conductive pathway”.
- 5) The authors detected a mid-gap band around -4.23 eV in the In:GaN and they assume that it could work as a channel to funnel electrons from the Ta₃N₅ to the collector given the experimental observation that the overall performance and “nbulk” improved. However, and here is where the relative thickness is important to know, it seems that the layer of In:GaN is relatively thin with respect Ta₃N₅, and in addition, there is no “electrical” proof that this states could act as a channel for the electrons.

I would recommend that the authors demonstrate first, using for instance, cyclic voltammetry that the In:GaN can transport charges at that “potential” or energy level. For instance, depositing a compact film of In:GaN on a conductive substrate and recording the cyclic voltammogram in the presence of an scavenger, could show whether the charges could flow directly from the substrate to the electrolyte through those mid-gap states.

Second, given the small thickness of the In:GaN deposited on the back contact, it seems plausible that the improvement observed comes from the “passivation” or smoothing of the original interface formed between Ta₃N₅ and Nb, rather than from an improved charge transport. In other words, given the architecture and the “distance” the electrons will travel through the In:GaN, it appears that this material will work as a buffer layer rather than as a scaffold or electron transporting material that accepts the electrons promoting the charge separation and collection.

- 6) The authors mentioned the stability and how it improved with these multijunction architecture, however it would be very appreciated if the authors analyze the sample after the stability test to identify what caused the drop of the performance. Was it just the electrocatalyst that detached? Did the authors find any degradation in the surface?

7) In page 15 the authors described the fitting of the EIS, but it is recommended to show the fitting in the plot as well to appreciate the quality of the model and the fitting.

8) In page 16, the authors perform a M-S analysis from what they claim to observed two distinct flat band potentials. This claim is quite misleading. By definition, the Ta₃N₅ will have a single flat band potential, the one that refers to the potential when the bands are flat. The explanation that starts from there is quite confusing, and I would recommend the authors to make an effort to clarify it, and if possible, to connect the explanation with the scheme show in the S18. Generally speaking, the flat band for those films corresponds to the values of 0.15 and 0.05 V respectively. Indeed, OCP measurements under illumination with high light intensity would corroborate that. The fact that the slope of the MS changes suggest that the band bending develops to a different extent in the different potential regions. At low applied potential, the existence of a certain degree of Fermi level pinning, caused by the existence a distribution of surface states in that potential region, caused the band bending to develop in a lesser extent while past of the applied potential drops at the Helmholtz layer. Likewise, at higher-potential the decrease of the density of surface states allows for a larger development of the band bending leading to a higher slope in the MS plot. Note that in the case of strong fermi level pinning the MS plot remains flat.

Reviewer #3 (Remarks to the Author):

In the manuscript “Interface engineering of Ta₃N₅ thin film photoanode for highly efficient photoelectrochemical water splitting”, the authors used interface engineering to improve the photon-to current efficiency to Ta₃N₅ photoelectrodes. Ta₃N₅ is interfaces with n-type In:GaN at the bottom and p-type Mg:GaN at the top using Nb as substrate. This work presents that the In:GaN layer mainly enhances the charge separation efficiency, which the Mg:GaN improves the surface charge injection efficiency. In general, I find the manuscript suitable for nature communication, but recommend that the following points be addressed before publication.

1. XRD only for the heterostructure on quartz glass is shown. However, the substrate material can have an impact on the conversion process/temperature. Therefore, XRD of the heterostructure on Nb foil should be provided, which represents the actual architecture of the investigated photoelectrode.

2. After conversion, the N/Ta ratio within the Ta₃N₅ films is about ~1.3 according to the AES profile (Fig. S3), which is significantly lower compared to previous studies and the ideal value of 1.67. Additionally, the Ga maps show the incorporation of Ga in the film and the absorption slight increases between 525 nm and 575 nm (Fig.S16). Lastly, the films show a strong PL emission at 2.6 eV (Fig. 3g). Typically, band-edge emission of Ta₃N₅ films is dominated by an emission peak at 2.1 eV (J. Mater. Chem. A, 2021,9, 20653-20663, Appl. Phys. Lett.,

2015, 107, 171902, ACS Catal., 2020, 10, 10316 —10324). Converting TaO to Ta₃N₅ sandwiched between In:GaN and Mg:GaN could have a significant impact on the material properties of the formed Ta₃N₅ film. What is the difference between the material properties (absorption, preferred orientation, composition, absorption etc) of the synthesized Ta₃N₅ films compared to previous reports? What is the impact of Ga-incorporation and the observed nitrogen deficiency on the band structure and PEC performance? How can the impact of modified bulk properties and improved interfaces be deconvoluted?

3. After annealing Fig. 3b shows the formation of an oxide-rich layer at the interface. Could this indicate that the interface layer is not completely converted to GaN in the heterostructure? Furthermore, from the AES depth profile (Fig. 1b vs Fig. 3b) it looks like that after deposition surface and back layer have a similar thickness, while after NH₃ annealing the surface layer appears to be reduced. Does the surface layer degrade during NH₃ annealing?

4. The thickness of the InO/GaO_x and TaO layer is reported after deposition. Similar information should be provided for the Mg-doped GaO_x layer.

5. The heterostructure was deposited on different substrates. The authors should state clearly on which substrates the measurements were conducted.

6. More experimental detail should be provided for the (crucial) NH₃ annealing step. This includes e.g. type of furnace, tube diameter, pressure, more specifics about the ramp up/ramp down rates/times.

Response to the reviewers' comments

Manuscript ID: NCOMMS-21-37650-T

Manuscript type: Research Article

Title: "Interface engineering of Ta₃N₅ thin film photoanode for highly efficient photoelectrochemical water splitting"

Correspondence Author: Yanbo Li

Authors: Jie Fu, Zeyu Fan, Mamiko Nakabayashi, Huanxin Ju, Nadiia Pastukhova, Yequan Xiao, Chao Feng, Naoya Shibata, Kazunari Domen

[The reviewer comments are shown in *italic*; responses are in blue; all revisions in the MS and SI are highlighted in red]

Reviewer #1

General comment: *This work by Li et al demonstrates a record-high applied bias photon-to-current efficiency of 3.46% for photoelectrochemical water splitting over Ta₃N₅-based photoanode. A n-i-p type heterostructure is formed by modifying the Ta₃N₅ bottom and top interfaces with n-type In-doped GaN and p-type Mg-GaN layers. The result is very exciting, especially the demonstration of effective electron and hole transport layers that potentially can be applied to other (oxy-)nitride semiconducting light absorbers. Overall, I find the paper to be well written and the characterization work also well carried out according to standard procedures. The effect of interface engineering is well worth publishing in Nature Communications. There are only a few questions the authors should clarify before acceptance.*

Response: We genuinely appreciate the reviewer for very positive evaluation of our work and for recommending our work for publication.

Comment 1: *In a previous work by the authors (Nature Catalysis, 3, 932-940, 2020), a Mg-doped Ta₃N₅ thin film was used as the light absorber to achieve an ABPE of 3.31%. The Mg-doped Ta₃N₅ seems to be more efficient than the pristine Ta₃N₅ used in this study. Why not using this type of Mg-doped Ta₃N₅ as the light absorber to construct the n-i-p heterostructure?*

Response 1: We thank the reviewer for this important question. In the initial stage of this project, we did try to use the Mg-doped Ta₃N₅ as the light absorber to construct the n-i-p heterojunction. However, the PEC performance of the Mg-doped Ta₃N₅ photoanode was deteriorated when the bottom interface was modified with In:GaN layer. We suspected that this was due to the diffusion of Mg from the Mg:Ta₃N₅ layer into the In:GaN layer. As Mg is a p-type dopant for GaN, this affects the n-type conductivity of the In:GaN layer. Thus, the electron extraction efficiency of the layer was lowered. Therefore, in this work we used pristine Ta₃N₅ to demonstrate the beneficial effect of the n-i-p heterojunction.

Comment 2: *For the synthesis of In-doped GaN layer, a InO_x-GaO_x bilayer was deposited by electron beam evaporation and ALD respectively as the precursor, while for the synthesis of Mg-doped GaN layer both MgO_x and GaO_x were deposited in a single ALD process. The latter seems to be better in achieving precursor films with more uniform doping. Why not using ALD for the*

deposition of InO_x precursor as well?

Response 2: Ideally, it is indeed better to use ALD for the doping of In into the GaO_x precursor film. Unfortunately, our ALD system only have limited precursor lines and In precursor is currently not available. However, due to the thin layer thickness (~5 nm) the high diffusivity of In (Solmi *et al.*, *J. Appl. Phys.* **2002**, 92, 1361) during the high-temperature nitridation process, successful doping of In into the GaN layer can be achieved with the InO_x - GaO_x bilayer precursor film.

Comment 3: The OER cocatalyst also plays a role in passivating surface defects in many photoanode. To better demonstrate the effectiveness of the surface passivation effect of the Mg:GaN layer and to show the effect of the cocatalyst, it is necessary to test the PEC performance of the photoanodes without cocatalyst modification.

Response 3: Thank the reviewer very much for raising this point. Following the reviewer's suggestion, we have tested the PEC performance of the pristine and interface-engineered photoanodes without co-catalyst modification. The data is provided as Supplementary Fig. 11 in the revised Supplementary Information of the manuscript. Without co-catalyst modification, the differences in the photocurrent and onset potential between the Ta_3N_5 and In:GaN/ Ta_3N_5 /Mg:GaN photoanodes are indeed larger, as compared to Fig. 4a.

Supplementary Fig. 11 | PEC performance of the Ta_3N_5 and In:GaN/ Ta_3N_5 /Mg:GaN photoanodes on Nb substrate without co-catalyst modification. The J-V curves were measured under chopped AM 1.5G illumination in 1 M KOH.

Reviewer #2

General comment: The manuscript “Interface Engineering of Ta_3N_5 thin film photoanode for highly efficient photoelectrochemical water splitting” describes that the simultaneous modification of Ta_3N_5 photoanodes with the deposition of Mg:GaN in the electrolyte interface and In:GaN in the back contact leads to a benchmark performance for solar water oxidation. The authors display good care when it comes to characterize the materials and interpret the results. However there some open questions that must be addressed in a major revision before further consideration of this paper for publication in *Nature Communications*.

Response: We thank the reviewer very much for offering us an opportunity to revise our manuscript

and for constructive comments which help to improve the quality of the paper. We have considered all suggestions and have modified the manuscript accordingly, including via the collection and inclusion of additional data. We sincerely hope that the details provided below will address the reviewer's concerns.

Comment 1: *In the introduction the authors motivate the importance of interface engineering invoking studies on perovskite solar cells. This is quite surprising given the large body of work on that particular area in the field of photoelectrochemical water splitting. I would encourage the authors to motivate the need for such “back” and “surface” engineering citing studies from the field of photoelectrochemistry since these align better with the topic of this study.*

Response 1: We thank the reviewer for this important suggestion. We wrote the introduction this way based on our previous research experience on perovskite solar cells. We agree with the reviewer that it is indeed better to motivate the need for interface engineering citing studies from the field of photoelectrochemistry. We have revised the introduction according to the reviewer's suggestion:

“Over the past decade, tremendous efforts have been devoted to interface engineering of various semiconductor photoelectrodes (e.g., Si,^{20,21} α -Fe₂O₃,²² BiVO₄,²³ Cu₂O²⁴) to improve their efficiency and stability through the passivation of interfacial defects or formation of hetero/homojunctions. Ideally, the semiconductor thin film light absorbers should be sandwiched by a n-type electron transport layer (ETL) and a p-type hole transport layer (HTL) to achieve efficient charge separation, similar to the “n-i-p” device architecture commonly used in thin-film photovoltaics²⁵. However, this type of “n-i-p” device architecture is rarely adopted in photoelectrochemical (PEC) devices²⁶, especially for some of the emerging PEC materials.”

20. Zhou, X. *et al.* Interface engineering of the photoelectrochemical performance of Ni-oxide-coated n-Si photoanodes by atomic-layer deposition of ultrathin films of cobalt oxide. *Energy Environ. Sci.* **8**, 2644-2649 (2015).
21. Moreno-Hernandez, I. A., Brunschwig, B. S. & Lewis N. S. Tin oxide as a protective heterojunction with silicon for efficient photoelectrochemical water oxidation in strongly acidic or alkaline electrolytes. *Adv. Energy Mater.* **8**, 1801155 (2018).
22. Zhang, H. *et al.* Gradient tantalum-doped hematite homojunction photoanode improves both photocurrents and turn-on voltage for solar water splitting. *Nat. Commun.* **11**, 4622 (2020).
23. Ye, K.H. *et al.* Enhancing photoelectrochemical water splitting by combining work function tuning and heterojunction engineering. *Nat. Commun.* **10**, 3687 (2019).
24. Li, C. Positive onset potential and stability of Cu₂O-based photocathodes in water splitting by atomic layer deposition of a Ga₂O₃ buffer layer. *Energy Environ. Sci.* **8**, 1493-1500 (2015).

Comment 2: *In page 4, the authors mentioned that “the full thermodynamic potential of Ta₃N₅ has yet to be unlocked”. I would recommend rephrasing this statement or dedicate some lines to explain what is considered “full thermodynamic potential”, perhaps the authors refer to the maximum photovoltage that could be delivered or the photocurrent, in such a case, it could be clearer to state those values to compare with the current state of the art.*

Response 2: We thank the reviewer for the constructive comment. The “full thermodynamic potential of Ta₃N₅ has yet to be unlocked” is mainly to point out the phenomenon that the onset potential of Ta₃N₅ photoanode achieved so far (usually >0.5 V vs. RHE) is far higher than its

theoretical limit (<0 V vs. RHE), which is the main reason why its actual PEC efficiency is still far lower than the theoretical efficiency. We have added the following sentence to explain this:

“Although photocurrent approaching its theoretical value has been achieved,¹⁴ the onset potential (usually >0.5 V vs. RHE) is still far higher than its theoretical limit (<0 V vs. RHE).”

Comment 3: *Although the authors show cross sectional figures, I could not find the thickness of the Mg:GaN, In:GaN or even Ta₃N₅. The authors need to include these values to better interpret and understand the results, as I will mention below.*

Response 3: Thank the reviewer very much for raising this point. The thickness of the Mg:GaN, In:GaN, and Ta₃N₅ layers are approximately 20, 5, and 600 nm, respectively. While the thickness of the Mg:GaN and Ta₃N₅ layers could be observed from the cross-sectional HRTEM results, the thickness of the In:GaN layer could not be directly measured and therefore was estimated from the number ALD cycles. We have added these values in our manuscript.

Comment 4: *Correct in page 9 “could severe as a conductive pathway” by “could serve as a conductive pathway”.*

Response 4: Thank the reviewer very much for pointing out this error. We have corrected this typo in our revised manuscript.

Comment 5: *The authors detected a mid-gap band around -4.23 eV in the In:GaN and they assume that it could work as a channel to funnel electrons from the Ta₃N₅ to the collector given the experimental observation that the overall performance and “bulk” improved. However, and here is where the relative thickness is important to know, it seems that the layer of In:GaN is relatively thin with respect Ta₃N₅, and in addition, there is no “electrical” proof that this states could act as a channel for the electrons. I would recommend that the authors demonstrate first, using for instance, cyclic voltammetry that the In:GaN can transport charges at that “potential” or energy level. For instance, depositing a compact film of In:GaN on a conductive substrate and recording the cyclic voltammogram in the presence of an scavenger, could show whether the charges could flow directly from the substrate to the electrolyte through those mid-gap states.*

Second, given the small thickness of the In:GaN deposited on the back contact, it seems plausible that the improvement observed comes from the “passivation” or smoothing of the original interface formed between Ta₃N₅ and Nb, rather than from an improved charge transport. In other words, given the architecture and the “distance” the electrons will travel through the In:GaN, it appears that this material will work as a buffer layer rather than as a scaffold or electron transporting material that accepts the electrons promoting the charge separation and collection.

Response 5: Thank the reviewer very much for the comments. Following the reviewer’s suggestion, we have prepared intrinsic and In-doped GaO_x films with a thickness of 200 nm by dual-source electron beam evaporation on Nb substrate and converted them into compact GaN and In:GaN films through nitridation. Then cyclic voltammetry of the samples was measured in 1 M KOH with 0.5 M H₂O₂ as a scavenger. The results in Supplementary Fig. 8a show that the electrons can be effectively injected from In:GaN layer to the electrolyte at a onset potential of 0.24 V vs. RHE, in contrast to a more cathodic value of -0.21 V vs. RHE for the GaN sample. The CV results and the corresponding energy diagrams suggest the In-induced inter-gap state can indeed act as a channel for the electrons.

Supplementary Fig. 8 | Electron transport properties of In:GaN and GaN films. The In:GaN and GaN compact films were obtained by nitriding In-doped GaO_x and pure GaO_x films deposited on Nb substrates through dual-source electron beam evaporation. **a**, Cyclic voltammetry of compact In:GaN and GaN films measured in 1 M KOH with 0.5 M H₂O₂ as a scavenger. **b**, Schematic energy diagram showing there is almost no barrier for the injection of electron from the Nb electrode through the In-induced inter-gap state to the electrolyte. **c**, UPS spectrum of GaN film deposited on Nb substrate. **d**, Schematic energy diagram showing there is a relatively high barrier for the injection of electron from the Nb electrode through the GaN layer to the electrolyte. These results verified that the In-induced inter-gap state can indeed act as a channel for electron transport through the In:GaN layer, resulting in the more positive onset potential and higher current value for the reduction current observed in **a**.

Regarding the roles of the In:GaN layer, we argued that the In:GaN layer not only facilitates electron transport but also passivates interface defects based on the PL and TRPL results in Fig. 3g-i. If the In:GaN layer only act as a defect passivation layer, then both the PL intensity and PL lifetime should increase after modification of the In:GaN layer. On the other hand, if the In:GaN layer only act as an electron transporting layer, then both the PL intensity and PL lifetime should decrease after modification of the In:GaN layer. The fact that the PL intensity decreases while the PL lifetime increases after modification of the In:GaN layer suggests both effects took place. Besides, the passivation effect is ascribed to the lattice matching between the GaN and Ta₃N₅ layers. If the In:GaN layer only act as a passivation layer, then sample with pure GaN underlayer would have similar PEC performance. However, as we show in Supplementary Fig. 17a, sample with pure GaN underlayer exhibits lower PEC performance. Therefore, we concluded that the In:GaN layer not only passivates interface defects but also facilitates electron transport.

Comment 6: The authors mentioned the stability and how it improved with these multijunction architecture, however it would be very appreciated if the authors analyze the sample after the stability test to identify what caused the drop of the performance. Was it just the electrocatalyst that detached? Did the authors find any degradation in the surface?

Response 6: Thank the reviewer very much for this important question. The NiCoFe-B_i OER co-catalyst that we recently developed (Feng *et al.*, *Nat. Commun.* **2021**, *12*, 5980) has excellent stability in alkaline condition as shown in Fig. R1. The co-catalyst is unlikely to degrade on the time scale (160 min) we tested for PEC stability. The decay of the photocurrent for the intrinsic Ta₃N₅ photoanode in Fig. 4d is likely due to degradation of the Ta₃N₅ surface due to self-oxidation. To confirm this, we have prolonged the stability test for the intrinsic Ta₃N₅ photoanode in Fig. 4d and characterized the change of surface composition before and after the PEC stability test by XPS. As shown in Supplementary Fig. 14, the oxygen content is significantly increased while the nitrogen content is decreased, which suggests the self-oxidation of the Ta₃N₅ surface during the PEC test. This finding is also consistent with previous understanding of the degradation mechanism of Ta₃N₅ based photoanodes (He *et al.*, *Chem* **2016**, *1*, 640-655).

Fig. R1 | Stability of NiCoFe-B_i co-catalyst deposited on FTO substrate for OER at 10 mA cm⁻² in 1 M KOH electrolyte.

Supplementary Fig. 14 | XPS spectra of Ta_3N_5 photoanode before and after PEC test. **a**, Survey spectra, and core-level spectra of **b**, $\text{Ta } 4f$, **c**, $\text{O } 1s$, and **d**, $\text{N } 1s$. The Ta_3N_5 photoanode modified with NiCoFe-B_i co-catalyst was tested in 1 M KOH at 1.0 V vs. RHE under AM 1.5G for 160 min. Afterwards, the NiCoFe-B_i co-catalyst on the surface was dissolved with diluted HCl for XPS characterization. The increased O-Ta-N peaks in **b** and Ta-O peak in **c** and decreased N $1s$ peak in **d** suggest the self-oxidation of the Ta_3N_5 surface, which accounts for the degradation of the photocurrent.

Comment 7: In page 15 the authors described the fitting of the EIS, but it is recommended to show the fitting in the plot as well to appreciate the quality of the model and the fitting.

Response 7: Thank the reviewer very much for the suggestion. To better show the fitting of the EIS data, we have changed the fitted line into red color in Fig. 5c. The EIS data can be well-fitted with the using the equivalent circuit models provided in Supplementary Fig. 20. We have added one sentence in Page 15 to appreciate the quality of the model and the fitting.

Comment 8: In page 16, the authors perform a M-S analysis from what they claim to observed two distinct flat band potentials. This claim is quite misleading. By definition, the Ta_3N_5 will have a single flat band potential, the one that refers to the potential when the bands are flat. The explanation that starts from there is quite confusing, and I would recommend the authors to make an effort to clarify it, and if possible, to connect the explanation with the scheme show in the S18. Generally speaking, the flat band for those films corresponds to the values of 0.15 and 0.05 V respectively. Indeed, OCP measurements under illumination with high light intensity would corroborate that. The fact that the slope of the M-S changes suggest that the band bending develops

to a different extent in the different potential regions. At low applied potential, the existence of a certain degree of Fermi level pinning, caused by the existence a distribution of surface states in that potential region, caused the band bending to develop in a lesser extent while past of the applied potential drops at the Helmholtz layer. Likewise, at higher-potential the decrease of the density of surface states allows for a larger development of the band bending leading to a higher slope in the MS plot. Note that in the case of strong fermi level pinning the MS plot remains flat.

Response 8: We thank the reviewer very much for this insightful comment which helps us better understanding the M-S data. We have revised the explanation for the observed two slopes in the M-S plots according to the reviewer's suggestion:

“The M-S plot of Ta₃N₅ photoanode in Fig. 5d shows that the flat-band potential is at 0.15 V vs RHE. However, the slope of the M-S plot changes at higher applied potential region (>0.77 V vs RHE), suggesting that the band bending develops to a different extent in different potential regions (Supplementary Fig. 18). At low potential region (<0.77 V vs RHE), the Fermi level pinning due to the existence a distribution of surface states causes the band bending to develop in a lesser extent while part of the applied potential drops at the Helmholtz layer⁴⁰. At high potential region (>0.77 V vs RHE), the decreased density of surface states allows for the band bending to develop in a higher extent, leading to a larger slope in the M-S plot. For the In:GaN/Ta₃N₅/Mg:GaN photoanode in Fig. 5e, the flat-band potential shifts cathodically to 0.05 V vs RHE while the transition of the two slopes happens at 0.37 V vs RHE. The lower applied potential to overcome the Fermi level pinning suggests there is a lower density of surface states due to the passivation effect of the Mg:GaN layer. The band bending can be developed in a larger extend at lower applied potential region, thus reducing the bias-potential requirement for the water oxidation process (Supplementary Fig. 18b).”

Reviewer #3:

Comments: *In the manuscript “Interface engineering of Ta₃N₅ thin film photoanode for highly efficient photoelectrochemical water splitting”, the authors used interface engineering to improve the photon-to-current efficiency to Ta₃N₅ photoelectrodes. Ta₃N₅ is interfaces with n-type In:GaN at the bottom and p-type Mg:GaN at the top using Nb as substrate. This work presents that the In:GaN layer mainly enhances the charge separation efficiency, which the Mg:GaN improves the surface charge injection efficiency. In general, I find the manuscript suitable for nature communication, but recommend that the following points be addressed before publication.*

Response: We greatly appreciate that the reviewer finds manuscript suitable for Nature Communication. We further would like to thank the reviewer for constructive comments which help to improve the quality of our article. The questions raised by the reviewer have also been fully addressed. We sincerely hope that the details provided below will address the reviewer's concerns.

Comment 1: *XRD only for the heterostructure on quartz glass is shown. However, the substrate material can have an impact on the conversion process/temperature. Therefore, XRD of the heterostructure on Nb foil should be provided, which represents the actual architecture of the investigated photoelectrode.*

Response 1: Thank the reviewer very much for this important question. According to the reviewer's suggestion, we have provided the XRD pattern of the InGaN/Ta₃N₅/Mg:GaN sample deposited on

Nb foil in Supplementary Fig 2c. We agree with the reviewer that the substrate may have an impact on the crystallization of Ta₃N₅ layer. However, comparing the Ta₃N₅ XRD patterns of the samples deposited on quartz glass and Nb substrates, we believe the substrates do not significantly affect the crystallinity of the Ta₃N₅ layer. The reason we use quartz glass substrate for XRD analysis is because the Nb substrate (forms Nb_xN after nitridation) exhibits many diffraction peaks overlapping with those of Ta₃N₅, which would interfere the analysis.

Supplementary Fig. 2c | XRD pattern of In:GaN/Ta₃N₅/Mg:GaN thin film on Nb substrate.

Comment 2: After conversion, the N/Ta ratio within the Ta₃N₅ films is about ~1.3 according to the AES profile (Fig. S3), which is significantly lower compared to previous studies and the ideal value of 1.67. Additionally, the Ga maps show the incorporation of Ga in the film and the absorption slight increases between 525 nm and 575 nm (Fig.S16). Lastly, the films show a strong PL emission at 2.6 eV (Fig. 3g). Typically, band-edge emission of Ta₃N₅ films is dominated by an emission peak at 2.1 eV (*J. Mater. Chem. A*, 2021,9, 20653-20663, *Appl. Phys. Lett.*, 2015, 107, 171902, *ACS Catal.*, 2020, 10, 10316-10324). Converting TaO to Ta₃N₅ sandwiched between In:GaN and Mg:GaN could have a significant impact on the material properties of the formed Ta₃N₅ film. What is the difference between the material properties (absorption, preferred orientation, composition, absorption etc) of the synthesized Ta₃N₅ films compared to previous reports? What is the impact of Ga-incorporation and the observed nitrogen deficiency on the band structure and PEC performance? How can the impact of modified bulk properties and improved interfaces be deconvoluted?

Response 2: Ideally, the N/Ta ratio in Ta₃N₅ should be 1.67. However, there exists a large amount of residual oxygen in Ta₃N₅ film converted from TaO_x precursor. In our previous study (Xiao *et al.*, *Nat. Catal.* **2020**, 3, 932-940) we revealed that the N/Ta atomic ratios are different on the surface and in the bulk of the Ta₃N₅ film (Fig. R2). The N/Ta ratio is indeed ~1.6 on the surface of the sample. However, the N/Ta ratio abruptly decreases to ~1.3 within the film. Therefore, the AES depth profile data is consistent with our XPS depth profile data reported previously.

Under high temperature nitridation process, it is indeed possible for the Ga to diffuse into the Ta_3N_5 layer. To investigate whether Ga incorporation in Ta_3N_5 could affect the absorption and PEC performance of Ta_3N_5 film, we prepared 1% Ga-doped TaO_x films (this doping concentration was selected by assuming all interface modified Ga are diffused into the Ta_3N_5 layer) by dual-source electron beam evaporation and converted them into Ga: Ta_3N_5 by nitridation. The absorption spectra in Fig. R3a reveal that there is no significant difference between the intrinsic and Ga-incorporated Ta_3N_5 films. Although the exact reason for the enhanced absorption in the 525-575 nm range requires further investigation, it is not due to Ga incorporation and may be caused by anti-reflection property of the multilayers. The PEC performance of the 1% Ga-doped Ta_3N_5 photoanode in Fig. R3b-c reveals that Ga-incorporation does not show a positive effect on improving the PEC efficiency of the photoanodes. Therefore, it is unlikely that the improved PEC performance after interface modification with In:GaN and Mg:GaN layers is due to the diffusion of Ga into the Ta_3N_5 film. The major difference between the material property of the synthesized Ta_3N_5 films in our study compared to previous reports is the lowered density of interface defects.

Regarding the PL emission of Ta_3N_5 , our previous study (Fu *et al.*, *ACS Catal.* **2020**, *10*, 10316-10324) shows that at room temperature the PL exhibits two emission peaks at 2.6 and 2.1 eV. This is ascribed to the strong optical anisotropy of its orthorhombic lattice that results in two intrinsic bandgaps of ~ 2.1 eV along the a-axis and ~ 2.6 eV along the b- and c-axes. This optical anisotropy has also been predicted by theoretical calculations (Fu *et al.*, *Appl. Phys. Lett.*, **2015**, *107*, 171902; Nurlaela *et al.*, *J. Solid State Chem.* **2015**, *229*, 219–227). At low temperature, we observed that the PL emission at 2.6 eV was significantly enhanced, while the emission at 2.1 eV was relatively weak. Recent study by Eichhorn *et al.* (*J. Mater. Chem. A* **2021**, *9*, 20653-20663) reveals that the 2.1 eV bandgap is indirect. Therefore, the PL emission at 2.1 eV requires the participation of phonons. At low temperature the phonon process is suppressed, resulting in the relatively weak PL emission at 2.1 eV.

Fig. R2 | The change of N/Ta atom ratio in pristine and gradient Mg: Ta_3N_5 films with Ar cluster etch time (Supplementary Fig. 12i from Xiao *et al.*, *Nat. Catal.* **2020**, *3*, 932-940).

Fig. R3 | a, Absorption spectra of intrinsic and 1% Ga-doped Ta₃N₅ films deposited on quartz glass. b, J-V curves for a batch of eight 1% Ga-doped Ta₃N₅ films deposited on Nb substrate. The photoanodes were modified with the NiCoFe-B_i co-catalyst and tested in 1 M KOH electrolyte under AM 1.5G illumination. c, Comparison of the ABPEs of intrinsic and 1% Ga-doped Ta₃N₅ based photoanodes.

Comment 3: After annealing Fig. 3b shows the formation of an oxide-rich layer at the interface. Could this indicate that the interface layer is not completely converted to GaN in the heterostructure? Furthermore, from the AES depth profile (Fig. 1b vs Fig. 3b) it looks like that after deposition surface and back layer have a similar thickness, while after NH₃ annealing the surface layer appears to be reduced. Does the surface layer degrade during NH₃ annealing?

Response 3: Thank the reviewer very much for raising this point. It is possible that there are some residual oxygen in the GaN layer after nitridation. However, the oxygen signal observed at the bottom interface in Supplementary Fig. 3b is more likely from the native SiO_x layer on the Si substrate. Comparing with the buried In:GaN layer, the top Mg:GaN layer is exposed to the NH₃ atmosphere. It is indeed plausible that the some of the Ga atoms can escape during the nitridation process. However, we note that the sputter depth shown in Supplementary Fig. 1b and 3b may not accurately reflect the layer thickness as the depth is converted from sputter time and different materials may have different sputter rates. Besides, the roughness of the sample also affects the resolution of the depth profile. Therefore, the AES depth profile is only used to show the existence of the multilayer structures before and after nitridation rather than the exact thicknesses of each layer.

Comment 4: The thickness of the InO/GaO_x and TaO layer is reported after deposition. Similar information should be provided for the Mg-doped GaO_x layer.

Response 4: Thank the reviewer very much for raising this point. The thickness of Mg-doped GaO_x layer made by ALD is approximately 20 nm. We have added this value in our manuscript in Page 19.

Comment 5: The heterostructure was deposited on different substrates. The authors should state clearly on which substrates the measurements were conducted.

Response 5: Thank the reviewer very much for the suggestion. We have clearly stated the type of substrates used for different measurements in the figure captions.

Comment 6: More experimental detail should be provided for the (crucial) NH₃ annealing step. This includes e.g. type of furnace, tube diameter, pressure, more specifics about the ramp up/ramp

down rates/times.

Response 6: Thank the reviewer very much for raising this point. The nitridation process was carried out in a horizontal quartz tube (inner diameter: 20 mm) furnace (MTI OTF-1200X). The quartz tube was purged with 100 sccm NH₃ (99.999%) flow at ambient pressure. The temperature was ramped at a rate of 10 °C min⁻¹ from room temperature to 1000 °C, maintained for 6 h, and cooled down naturally to room temperature. We have added these experimental details in our manuscript in Page 19.

REVIEWERS' COMMENTS

Reviewer #1 (Remarks to the Author):

After reading the revised manuscript, I think the concerns from me have been satisfactorily addressed. Considering the significance of this study in the arrange of Ta₃N₅ thin film photoanode, I would like to recommend its publication now.

Reviewer #2 (Remarks to the Author):

The authors have addressed all the concerns I posed on previous revision and, with the complementary information and experiments they provided, I consider that the study matches the standards of Nat. commun. and should be accepted for publication without further revision.

Reviewer #3 (Remarks to the Author):

The authors addressed the comments sufficiently. I recommend the paper “Interface Engineering of Ta₃N₅ thin film photoanode for highly efficient photoelectrochemical water splitting” for publication in Nature Communication.